# To Be Active or to Stop? A Cross-Sectional Retrospective Study Exploring Provider Advice and Patient Fears Surrounding Physical Activity in Pregnancies Complicated by Fetal Growth Restriction

**DOI:** 10.3390/ijerph19106076

**Published:** 2022-05-17

**Authors:** Rachel A. Tinius, Jill M. Maples, Mark A. Schafer, Alissa Paudel, Kimberly B. Fortner, Nikki B. Zite, Taniya S. Nagpal

**Affiliations:** 1School of Kinesiology, Recreation, and Sport, Western Kentucky University, Bowling Green, KY 42014, USA; mark.schafer@wku.edu; 2Department of Obstetrics and Gynecology, University of Tennessee Graduate School of Medicine, Knoxville, TN 37996, USA; jmaples1@utmck.edu (J.M.M.); apaudel@utmck.edu (A.P.); kfortner@utmck.edu (K.B.F.); nzite@utmck.edu (N.B.Z.); 3Department of Kinesiology, Faculty of Applied Health Sciences, Brock University, St. Catharines, ON L2S 3A1, Canada; tnagpal@brocku.ca

**Keywords:** exercise, pregnant, intrauterine growth restriction, guidelines

## Abstract

Exercise guidance for women with pregnancies complicated by fetal growth restriction (FGR) is vague, despite the fact that physical activity during pregnancy improves placental development, placental blood flow, and encourages healthy fetal growth. The goal of this study is to test the hypothesis that women with pregnancies complicated by FGR are fearful of physical activity and are being given unclear or limited advice from healthcare providers. Participants (N = 78) (women who delivered an infant diagnosed with FGR within the past 5 years) took an electronic survey including demographic information, pregnancy information, provider advice recall, and other health-related information relevant to growth-restricted pregnancies. Quantitative and qualitative (post-positivism paradigm) methods were employed to analyze the data. When asked specifically about how the FGR diagnosis impacted their activity levels, nearly 50% of participants said the diagnosis led them to decrease their activity levels. Participants reported that healthcare providers who do discuss activity with pregnant patients with FGR suggest low-intensity activities or ceasing activity, although the majority of providers did not discuss activity at all. More fears surrounding physical activity were reported post-FGR diagnosis, including worrying about fetal growth and development and causing fetal harm when engaging in physical activity.

## 1. Introduction

Fetal growth is an important predictor of pregnancy and neonatal outcomes and is often a reflection of physiological and pathological factors influencing the fetus [1]. Fetal growth restriction (FGR) is considered “the most common and complex problem in modern obstetrics” by the American College of Obstetrics and Gynecology [2]. Consensus-based definitions for early and late FGR include specific combinations of several parameters, including estimated fetal weight (EFW) (<3rd centile), fetal abdominal circumference (AC) (<3rd centile), or EFW or AC (<10th centile) with sonographic findings indicative of insufficient placental and fetal blood flow [3]. Patients with an FGR diagnosis are often referred to a Maternal–Fetal Medicine Specialist for additional monitoring [4] because FGR is associated with fetal and neonatal mortality and morbidity [1].

Preventative and treatment options for FGR are limited [5]. Current management consists of serial fetal growth ultrasounds, assessment of fetal wellbeing, and placental blood flow (via Doppler ultrasound), all of which may be used to determine the most appropriate timing for the delivery of a growth-restricted fetus [2,5,6].

For pregnancies complicated by FGR, there is a lack of consensus regarding physical activity recommendations. Physical activity is a potent vasodilator, which increases placental angiogenesis and improves endothelial function during pregnancy [7,8]. Additionally, physical activity has been associated with appropriate fetal growth and positive maternal and neonatal outcomes in uncomplicated pregnancies [9,10]. The majority of prenatal physical activity guidelines recommend that moderate-intensity exercise be avoided during pregnancy complicated by FGR [11]. The American College of Obstetricians and Gynecologists does not include FGR as an absolute contraindication to exercise during pregnancy [12], but provides limited guidance in terms of exercise counseling for pregnant patients with FGR. Bed rest is not recommended to treat FGR and activities of daily living should be continued according to the Society for Maternal Fetal Medicine [5]. However, little is known about how pregnant patients with FGR may be changing their physical activity patterns in response to an FGR diagnosis and whether healthcare providers are talking to their patients about physical activity after a diagnosis of FGR.

An important factor to consider when examining maternal physical activity patterns and patient–provider communication is fear and anxiety associated with an FGR diagnosis. Nearly one out of every four pregnancies is deemed “high-risk” for various reasons, and unfortunately, this classification appears to exacerbate anxiety and fears that for many women are already a normal part of pregnancy [13]. Fairbrother et al. found that women with high-risk pregnancies had a 5.2× greater incidence of anxiety compared with women with low-risk pregnancies [14]. Furthermore, women with anxiety and/or depression during pregnancy are more than twice as likely to be hypertensive during pregnancy when compared with women without anxiety and/or depression [15]. Given the strong connection between maternal physical and mental health during pregnancy [16], as well as the well-established connection between maternal blood pressure and FGR [17], it is important to consider the implications of maternal fear and anxiety on behavior and ultimately fetal/infant health. It is plausible that fears and anxiety surrounding an FGR diagnosis may further perpetuate a decrease in physical activity, which could have important negative implications for fetal growth.

This study proposes an important first step to propel further investigation into the role of physical activity in FGR pregnancies. The aim of this research was to understand physical activity practices and feelings surrounding physical activity before and after an FGR diagnosis, as well as to understand obstetric provider recommendations about physical activity before and after FGR diagnosis, as understood by women with a history of an FGR pregnancy. We hypothesize that pregnant women may have increased fears about physical activity during their pregnancy and ultimately reduce physical activity levels after the FGR diagnosis. Similarly, we suspect provider communication on the topic of physical activity during pregnancies complicated by FGR is limited or confusing.

## 2. Materials and Methods

### 2.1. Participants

Women were recruited via FGR social media support groups. Inclusion criteria consisted of the delivery of a baby with known or suspected FGR over the previous five years. Pregnancy is considered to be a very salient time in a woman’s life; as such, research suggests that women recall details from their pregnancy very well, even up to 10–15 years later [18].

### 2.2. Procedures

All data were captured electronically through a survey generated by our team and distributed via Research Electronic Data Capture (REDCap) [19] and in accordance with STROBE guidelines for observational studies [20]. Once a participant clicked the link, they were given the consent form and an electronic signature was obtained. Once they consented to participation, they were given access to the survey. They were asked to complete the survey in one sitting, if possible. However, they were allowed to save and return if needed. The survey included demographic information, pregnancy information, provider advice information, and other health-related information that is relevant to growth-restricted pregnancies (e.g., stress, drug use, and medication use). All questions were adapted from validated tools, when possible, and all survey questions were carefully developed by our team of researchers and clinicians. The exercise during pregnancy information was adapted from the American College of Sport’s Medicine’s Exercise Vital Signs [21], which consists of two open-ended questions:On average, how many days per week do you engage in moderate-to-vigorous physical activity (such as a brisk walk)?On average, how many minutes do you engage in physical activity at this level?

The number of days and number of minutes were then multiplied to determine the number of minutes per week. The number of minutes per week was determined for each trimester as exercise levels tend to decrease throughout pregnancy [22].

### 2.3. Data Analysis

Quantitative data analyses were conducted using SPSS Statistics, Version 28 (IBM Corp., Armonk, NY, USA). Shapiro–Wilk tests were conducted to determine normality of the data. For continuous variables, means and standard deviations were calculated and for categorical variables, counts and percentages were determined. A repeated-measures ANOVA was run to examine changes in PA levels over time among participants. A one-sample t-test was used to determine whether physical activity levels at each time point were significantly different from the recommended level (150 min/week). In order to determine relationships between variables, Spearman correlation coefficients were utilized.

To evaluate the open-ended responses, a post-positivism paradigm was used, which applies objectivity to qualitative data such that findings can be categorized and quantified, indicating the number of individuals who are represented for each theme or response [23,24]. As such, to assess the open-ended questions we completed a content analysis following the methodology of Powers and Knapp (2006) to provide frequency counts for the responses [25]. TSN read and categorized the data for each open-ended question into themes that would summarize the answers, and then the number of individuals (frequency) who indicated that response was calculated as a percentage. Content analysis findings were then reviewed and confirmed by two independent researchers (RT and JM). Of note, content analyses findings following this methodology are meant to be exploratory and inferential analyses, such as trying to identify significant differences in frequency of responses, are not recommended or cannot be used to formulate a conclusion [23,24,25]. Therefore, all content analyses are exploratory to describe the data. Questions evaluated by content analysis were pregnancy complications, physical activity advice given by healthcare providers before and after the FGR diagnosis, and fears related to physical activity pregnant women had before and after the FGR diagnosis.

## 3. Results

A total of 78 women participated in the study. Demographic characteristics and pregnancy complications are shown in Table 1 and Table 2.

### 3.1. Provider Conversations Regarding Physical Activity during Pregnancy

Prior to diagnosis of FGR, 31 women (38.8%) reported that their doctor(s) discussed physical activity during pregnancy with them. After diagnosis of FGR, only 16 (20.5%) reported a conversation regarding physical activity (and how the diagnosis would/wouldn’t change the recommendations). Taken together, there was a significant decrease in provider communication about physical activity after the FGR diagnosis (*p* = 0.044).

### 3.2. Physical Activity Levels

PA levels (total minutes per week) went down among all women from pre-pregnancy to the third trimester (*p* < 0.001) (Figure 1), and the decrease in activity was significant from each timepoint to the next (pre-pregnancy to first trimester, *p* < 0.001, first trimester to second trimester, *p* = 0.024, second trimester to third trimester, *p* = 0.001). Women went from an average of 142 min per week of self-reported PA to 38 min per week, which is significantly below ACOG recommendations for women during pregnancy (150 min per week) (*p* < 0.001). When asked specifically about how the FGR diagnosis impacted their activity levels, nearly 50% of the women said the diagnosis led them to decrease PA levels.

Physical activity levels decreased throughout the pregnancy to well below the established recommendations for pregnancy.

There were no relationships noted between estimated fetal weight (during pregnancy) and PA levels in each trimester (Table 3). A positive relationship was found between physical activity (second trimester only) and birthweight.

### 3.3. Content Analysis

Before being diagnosed with FGR, 38.8% of participants indicated that their provider discussed physical activity with them. Content analysis showed that most providers suggested exercises such as walking, yoga, light-intensity workouts, and generally recommended continuing doing activities that were performed before pregnancy. For example, participants were told “*To limit extreme exercise/heart rate, but that walking & small workouts were ok*” and “*Continue to exercise as a pregnant woman would.*” Additionally, some healthcare providers specified to be active for at least 30 min every day or generally to try and stay active throughout the day without a specific time recommendation.

After being diagnosed with FGR, 20.5% of participants indicated that their provider discussed physical activity with them. The content analysis showed that providers predominantly suggested that participants reduce intensity, engage in light-intensity exercises such as walking, continue doing what was done before they were pregnant or do no exercise at all. For example, “*Don’t get too winded, exercise at an easy intensity*” or “*Daily movement was encouraged, but not excessive or intense exercise.*” In regard to length of exercise, participants were told to either engage in bedrest, 30 min of daily activity, or generally stay active throughout the day. Of note, nearly 80% of women reported no physical activity advice from their healthcare provider after the FGR diagnosis. As one woman stated, “*and the only true advice that my physician gave me was to increase the amount of food I was eating. He stated that I need to be consuming 2400 cal a day to help baby grow. He never mentioned exercise, never mentioned water intake, nothing like that. So of course I assumed I need to be on ‘light duty’ per se and focused on stuffing my face. And not even good foods.*”.

Respondents who provided examples of fears surrounding physical activity before their FGR diagnosis highlighted generally wanting to avoid harming the fetus or maternal well-being, bleeding, and miscarriage. For example, participants shared general fears, such as “*Over extension could harm the pregnancy,*” and “*I just didn’t know how much I could exercise or if it would possibly hurt myself or my baby*”. More fears surrounding physical activity were reported post-FGR diagnosis. Fears surrounding fetal growth and development and generally causing fetal harm were described, including specific concerns for restricting nutrient transport by means of weight loss or burning too many calories when engaging in physical activity. Participant quotes regarding fetal harm projected the notion that they feared *“making it worse”* or causing additional harm, for example “*I worked a very fast paced physical job and was worried that the activity was depleting nutrients from my unborn baby and potentially further complicating the IUGR diagnosis*” and “*That I would continue hurting her*”. Additionally, participants highlighted worrying about stillbirths or miscarriages, increasing existing complications such as hypertension and general safety concerns. When assessing other pregnancy complications, most participants indicated feeling mental health constraints as a result of the FGR diagnosis such as heightening stress and anxiety. For example, “*After my IUGR diagnosis, I started to develop high blood pressure due to anxiety and stress,*” and “*The stress/anxiety/depression was mostly related to the diagnosis of IUGR.*” Figure 2 depicts content analysis findings with corresponding frequencies for each response.

Prior to FGR diagnosis, women reported fewer fears and more PA advice regarding physical activity when compared with after the FGR diagnosis.

## 4. Discussion

Physical activity levels decreased throughout pregnancy among a sample of women with FGR, and nearly 50% of them reported that the decrease occurred after their diagnosis of FGR. These data suggest that there are concerns and/or misunderstandings surrounding the role of physical activity on FGR pregnancies, which is corroborated by the fact that 30 women reported fears about physical activity after FGR diagnosis, compared with only 8 women prior to having FGR diagnosed. Furthermore, the percentage of women who reported conversations about physical activity during pregnancy with their provider was exceedingly low after the diagnosis of FGR (20%), which supports the notion that physical activity is either not being discussed or is even discouraged (e.g., with bedrest (Figure 2)) among women with an FGR diagnosis, which is in contrast to recommendations set forth by the Society for Maternal and Fetal Medicine [5].

Our data suggest that providers are not discussing physical activity with the majority of patients, particularly after a diagnosis of FGR. Our finding is consistent with existing work suggesting healthcare providers are responsible for providing prenatal PA advice and counseling; however, barriers exist causing them to seldom report performing this role for pregnant patients [26]. We suspect that this lack of communication on the topic may be related to the fact that limited guidance exists even from key pregnancy constituents such as ACOG. Data on the impacts (positive or negative) of physical activity in a pregnancy complicated by FGR are extremely limited, posing a significant challenge to obstetric providers attempting to make specific physical activity recommendations for high-risk pregnancies. A recent review states that the risks of moderate-to-vigorous physical activity outweigh the benefits in FGR pregnancies, and that women with FGR should avoid exercise [11]. However, the collective data (albeit limited) suggest physical activity could be safe or even beneficial for pregnancies complicated by FGR. For example, Clapp et al. concluded that the maintenance of a low-volume physical activity regimen throughout pregnancy (in a non-FGR pregnancy) can actually stimulate fetoplacental growth [27]. The exact role of physical activity on fetal growth in a pregnancy complicated by FGR has not been investigated, and is an important future area for research. It is possible that the recommendation to reduce or limit physical activity for pregnancies complicated by FGR, in an effort to be cautious, could actually further perpetuate growth restriction.

A recent commentary took the position that contraindications to exercise during pregnancy may actually be restricting further investigation into the potential beneficial or protective role physical activity may have, even for complicated pregnancies [28]. Specifically in relation to FGR, the main rationale that has led to restricting activity is the notion of reducing blood flow and thus adding further detriment to fetal development. However, physical activity is a well-established potent vasodilator, which can enhance placental angiogenesis, and improve endothelial function during pregnancy [7,8,29]. Bauer et al. stated that “*With regard to the systolic/diastolic ratio (S/D ratio)—a standard method for measuring the function of the umbilical cord artery—a number of research results confirm that physical activity increases the umbilical blood flow and improves placental circulation*” [29]. Therefore, it seems that restricting or eliminating physical activity could, in fact, be counterproductive to some pregnancies with an FGR diagnosis, depending on the severity of blood flow impedance along the fetal component of the placental unit. Underscored by our findings, healthcare providers may be choosing to err on the side of caution by suggesting limiting physical activity, which could be unknowingly denying several maternal and fetal health benefits.

Another important factor to consider when discussing physical activity in pregnancies complicated by FGR is that this is not a homogenous group. For example, there are a wide array of other health concerns that can complicate/contribute to the severity of FGR diagnoses (Table 2). Additionally, the causes of FGR are diverse, ranging from fetal, maternal, and uterine/placental to demographic factors [30,31]. Also, many patients diagnosed with FGR will ultimately deliver a constitutionally small but otherwise healthy baby. For these and likely other reasons, future recommendations for physical activity during a pregnancy complicated by FGR may have to be further subdivided based on other factors such as findings from routine biweekly ultrasounds (e.g., FGR pregnancies with absent or reversed-end diastolic velocities may need recommendations very different from those for FGR pregnancies with normal Doppler assessments, and FGR pregnancies with diagnosed placental insufficiencies may be managed differently than those without).

Consistent with the fact that providers are not counselling patients on PA post FGR diagnosis, PA levels significantly decreased post FGR diagnosis in the recall of the present study cohort. Generally, PA levels tend to decrease throughout pregnancy [32], so it is difficult to discern changes due to normal adaptations in pregnancy physiology vs. uncertainty and fear surrounding FGR diagnosis. However, nearly 50% of the study cohort reported reducing their physical activity levels after the FGR diagnosis. Clinical guidelines suggest that physical activity should be maintained throughout pregnancy if possible, to maximize the positive benefits for mother and baby [12,33]. Given that women tend to become less active and do not meet activity recommendations [32] (which is already a concerning obstetric issue), an FGR diagnosis may perpetuate inactivity among women who may have much to gain from staying active.

No relationships were found between estimated fetal weight (pre-delivery) and physical activity levels in each trimester, suggesting that physical activity is, at least, not harmful to infant growth, which is consistent with previous work [34]. Our study did note a positive correlation between physical activity during the second trimester and infant birthweight, which is reinforced by another recent study suggesting that higher levels physical activity are associated with a higher infant birth weight and decreased rates of small for gestation age infants (SGA) [35].

Notably, women report more fears about physical activity after a diagnosis of FGR, which creates even more of a need to support and encourage women to stay active during a growth-restricted pregnancy. Based on the specific fears reported in open-ended responses, it is clear that women with an FGR diagnosis were concerned that physical activity could cause issues with the growth of their baby, as only one woman reported this as a fear prior to diagnosis, yet eleven did after.

The fears reported in the present analysis of post-FGR diagnosis are concerning, as the majority of the respondents indicated a form of ‘self-blame’ for causing harm to their developing fetus. Comments such as “continuing to harm her” or using terms such as “Because of me” suggest that women are internalizing responsibility for FGR, and this could be further exacerbating fears for engaging in a healthful behavior such as physical activity. This is problematic as previous studies have reported that during pregnancy women can engage in self-blame for prenatal complications, even when outside of maternal control (i.e., based on biological or environmental factors) [36,37,38]. In fact, one qualitative examination noted that women who have been prescribed bed rest report increased loss of self-control over their bodies and feeling like a ‘failure’ [39]. Engaging in self-blame and consequently experiencing stress throughout gestation can heighten the risk for post-partum mental health complications [36,38,40], as well as impact perinatal outcomes such as gestational hypertension and preeclampsia [41]. Prior to the FGR diagnosis, the fears shared by this cohort are quite commonly reported in the literature. These fears include being unaware of how to be active and generally questioning safety [42,43,44], whereas post-diagnosis physical activity was being questioned as a potential blockade to the transport of fetal nutrients and damaging fetal health. These findings emphasize the critical need to better translate safety-related information on maternal physical activity, which includes debunking myths that associate activity with undue fetal harm and supporting women with the prevention of self-blame.

To our knowledge, this is the first study to explore physical activity levels, advice on activity received from healthcare providers, and fears surrounding activity before and after an FGR diagnosis. The strengths of this study include the large sample assessed, and the collection of both quantitative and open-ended responses to explore our research objectives, providing more detail on the lived experience of FGR and the relationship with physical activity. However, as this was an online survey, limitations include the potential risk of bias (selection and recall) and the use of self-reported data. As noted, limited research has investigated the potential effects of prenatal physical activity in FGR pregnancies despite limited evidence to suggest that activity should be completely restricted, and contrarily, ample evidence supporting enhanced placental function in active pregnancies. As such, the exploratory nature of this study is a first step to informing the need for prospective investigations, which can also include objective measures of physical activity in FGR pregnancies and future intervention research.

## 5. Conclusions

In conclusion, a diagnosis of FGR in pregnancy appears to reduce maternal activity levels and elicit fears surrounding activity stunting fetal development. Women in this study report that many healthcare providers who do discuss activity with pregnant patients with FGR suggest low-intensity activities or ceasing activity, although a majority of providers did not discuss activity at all. Given the documented angiogenic benefits of physical activity in pregnancy, further research is needed in pregnant populations diagnosed with FGR to investigate potential benefits for maternal and fetal well-being. In addition, health promotion initiatives for prenatal physical activity should consider messaging that would prevent maternal self-blame for prenatal complications such as FGR, and instead, push the focus on safe exercises that could be performed to support women in overcoming the fears and uncertainties associated with being active.

## Figures and Tables

**Figure 1 ijerph-19-06076-f001:**
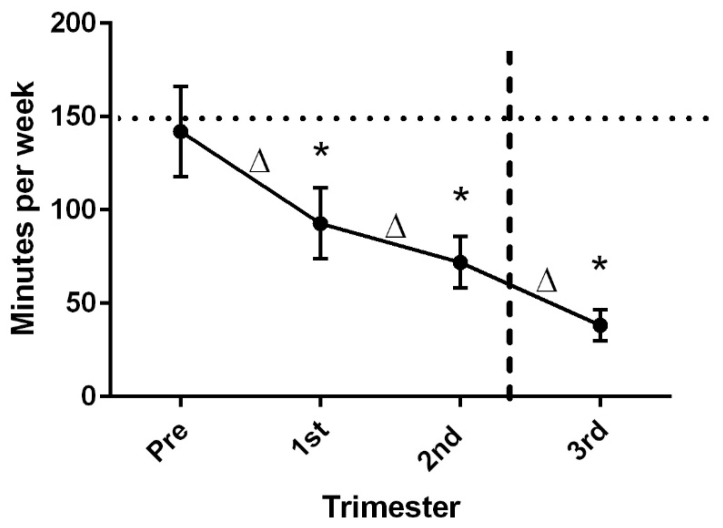
Physical activity levels in pregnancies complicated by FGR. Note: 1st, 2nd, and 3rd trimester timepoints on the graph represent the middle of each trimester. * *p* < 0.05, difference in PA level from recommended level (150 min/week); Δ < 0.05, change in PA from one timepoint to the next; ···· denotes recommended amount of physical activity before and during pregnancy (150 min/week); ---- denotes mean timepoint of FGR diagnosis (25.5 ± 5.9 weeks gestation).

**Figure 2 ijerph-19-06076-f002:**
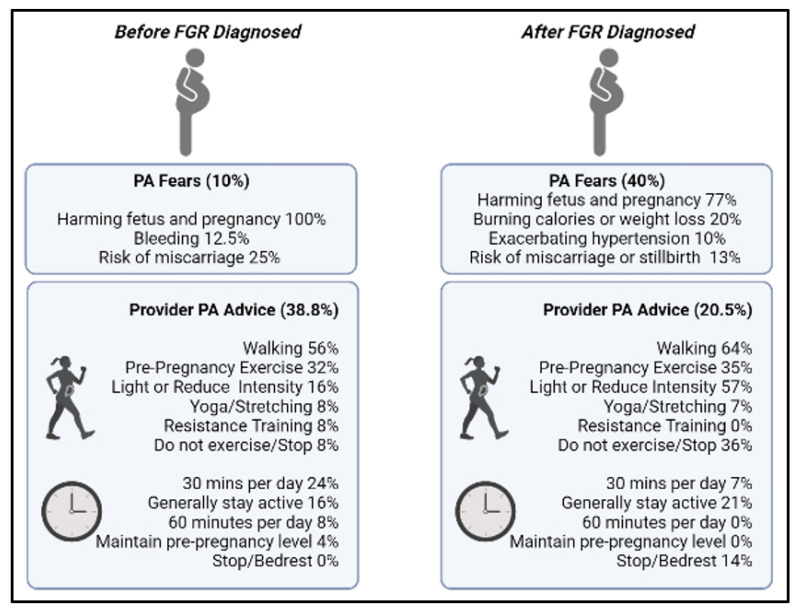
Summary of content analysis findings highlighting physical activity fears and healthcare provider advice for physical activity before and after FGR diagnosis. Nonbold text and percentages refer to respondents among those who indicated ‘yes’ to the variable (i.e., fears and receiving advice) and provided an example. FGR—fetal growth restriction; PA—physical activity; bold text and percentages refer to full sample. This figure was created with BioRender.com (accessed on 8 April 2022).

**Table 1 ijerph-19-06076-t001:** Demographic characteristics.

Characteristic (N = 78)	Mean ± SD or N (%)
Age (years)	31.5 ± 5.0 (range 18–41)
Pre-pregnancy weight (lbs)	157.9 ± 48.7
Pre-pregnancy height (inches)	64.0 ± 4.7
Pre-pregnancy body mass index (kg/m^2^)	26.8 ± 7.7
Time of FGR diagnosis (gestation age in weeks)	25.5 ± 5.9
Parity	
Nulliparous	38 (48.7)
Multiparous	40 (51.2)
Race	
White	54 (69.2)
Black	3 (3.8)
Asian	3 (3.8)
Missing	18 (23.0)
Ethnicity	
Hispanic, Latino, Spanish origin	2 (2.6)
Non-Hispanic	59 (75.6)
Missing	17 (21.8)
Educational Attainment	
No school	1 (1.3)
High school diploma	7 (9.0)
GED	2 (2.6)
Some college credit	16 (20.5)
Associate’s degree	4 (5.1)
Bachelor’s degree	17 (21.8)
Master’s degree	12 (15.4)
Professional degree	1 (1.3)
Doctorate degree	2 (2.6)
Missing	16 (20.5)
Gestation Age at Delivery (weeks)	35.1 ± 4.8
Infant Birthweight (g)	1834 ± 752
Infant Percentile at Birth (%)	
<3%	48 (61.5)
3–10%	15 (19.2)
>10%	9 (11.5)
UTD	6 (7.7)

UTD: unable to determine (i.e., missing weight, gender, or GA at delivery).

**Table 2 ijerph-19-06076-t002:** List of pregnancy and fetal complications in conjunction with FGR.

**Pregnancy Complication ***	**N (%)**
High blood pressure	25 (32.1)
Oligohydraminos	20 (25.6)
Anxiety/stress/depression	18 (23.1)
Group B Strep	11 (14.1)
Gestational diabetes	10 (12.8)
Preterm labor	2 (2.6)
Hyperemesis gravidarium	1(1.3)
Fatigue	1 (1.3)
Pain—low back, ankles	1 (1.3)
Shingles	1 (1.3)
**Fetal Complication** ^ **#** ^	**N (%)**
Reverse fetal Doppler	8 (10.3)
Absent fetal Doppler	14 (17.9)
Abnormal BPP	12 (15.4)
Abnormal NST	11 (14.1)

BPP—biophysical profile, NST—non-stress test; * note: 46 women (56%) reported at least one pregnancy complication, 29 women reported 2+ complications; ^#^ note: 13 women (16.7%) reported at least one fetal complication, 13 women reported 2+ complications.

**Table 3 ijerph-19-06076-t003:** Relationships between birthweight, estimated fetal weight, and physical activity levels during by trimester.

Physical Activity Level (min/Week)	Estimated Fetal Weight (%) (Mean: 13.6 ± 27.7%)	Birthweight (Mean: 4.0 ± 1.3 lbs)
1st Trimester	R = −0.148, *p* = 0.232	R = 0.145, *p* = 0.252
2nd Trimester	R = −0.056, *p* = 0.651	R = 0.266, *p* = 0.023 *
3rd Trimester	R = −0.093, *p* = 0.470	R = 0.179, *p* = 0.157

Relationship between EFW and BW (r = 0.370, *p* < 0.001). * *p* < 0.05.

## Data Availability

The datasets used and/or analyzed during the current study are available from the corresponding author on reasonable request.

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
