# Peer review of "To Be Active or to Stop? A Cross-Sectional Retrospective Study Exploring Provider Advice and Patient Fears Surrounding Physical Activity in Pregnancies Complicated by Fetal Growth Restriction"

_ijerph, 2022, doi:10.3390/ijerph19106076_

Round 1

Reviewer 1 Report

The discussed problem of low birth weight and physical activity in pregnancies is crucial, has a meaningful impact on maternal and children’s health, and requires further analysis. I am pleased to have the possibility to review the study “To be active or to stop? Exploring provider advice and patient fears surrounding physical activity in pregnancies complicated by fetal growth restriction”. The analysis was performed very well according to STROBE guidelines for observational studies. I would suggest mentioning it in your methodology with a citation of it.

The size of the population, the character of the study and the methodology are undoubtfully strengths of the study.

As a minor point, I would suggest underlining the aim of the study at the end of the introduction and making it more transparent. I also disagree with you about the problematic definition of FGR (lines 34-35). I strongly suggest performing further literature research looking for definitions made by ACOG, RCOG and ISUOG. In lines 40-41 is written “…Placental blood flow, monitored by Doppler ultrasound, is recommended for fetal surveillance and is the primary assessment…” – is not entirely correct. You used a shortcut. I recommend making the sentence clear for the readers and paraphrasing it. Both tables 1 and 2 were done for the whole population. I recommend presenting differences across the study and control group statistically instead.

Reviewer 2 Report

This is a very interesting manuscript about the relationship between physical activity and fetal growth restriction or small fetuses for gestational age. Despite many limitation factors, it is very well written, and the conclusions are supported by the objectives.

However ,  I have the following observations in order to improve it :  

Line 34-35 : fetal growth restriction is a very well defined -see Delphy consensus procedure::

 Gordijn SJ, Beune IM, Thilaganathan B, et al. . Consensus definition of fetal growth restriction: a Delphi procedure. Ultrasound Obstet Gynecol 2016;48:333–9. 10.1002/uog.15884 

Line 43: There is prevention for the high-risk population of FGR after the first-trimester ultrasound using Aspirin, please correct this.

Line 137-142: Please explain the context to discuss physical activity in an obstetric visit? From table 1 , I understand that your analysis has included women with a BMI of 26? What is their status on the physical activity before pregnancy?
